# EZNAS: Evolving Zero-Cost Proxies For Neural Architecture Scoring

**Yash Akhauri**[1]    **J. Pablo Muñoz**[2]    **Nilesh Jain**[2]    **Ravi Iyer**[2]
[1]Cornell University    [2]Intel Labs
ya255@cornell.edu
{pablo.munoz,nilesh.jain,ravishanker.iyer}@intel.com

## Abstract

Neural Architecture Search (NAS) has significantly improved productivity in the design and deployment of neural networks (NN). As NAS typically evaluates multiple models by training them partially or completely, the improved productivity comes at the cost of significant carbon footprint. To alleviate this expensive training routine, zero-shot/cost proxies analyze an NN at initialization to generate a score, which correlates highly with its true accuracy. Zero-cost proxies are currently designed by experts conducting multiple cycles of empirical testing on possible algorithms, datasets, and neural architecture design spaces. This experimentation lowers productivity and is an unsustainable approach towards zero-cost proxy design as deep learning use-cases diversify in nature. Additionally, existing zero-cost proxies fail to generalize across neural architecture design spaces. In this paper, we propose a genetic programming framework to automate the discovery of zero-cost proxies for neural architecture scoring. Our methodology efficiently discovers an interpretable and generalizable zero-cost proxy that gives state of the art score-accuracy correlation on all datasets and search spaces of NASBench-201 and Network Design Spaces (NDS). We believe that this research indicates a promising direction towards automatically discovering zero-cost proxies that can work across network architecture design spaces, datasets, and tasks.

## 1   Introduction

The manual trial-and-error method of designing neural network architectures and assessing whether it meets performance and accuracy requirements is not scalable to complex neural architecture design spaces and hardware deployment scenarios. Neural Architecture Search is intended to address this problem of iterative and expensive design by automating the search of neural network architectures. Enabling such automation requires accurate and efficient prediction of the accuracy of candidate neural network architectures. The process of predicting the accuracy often involves partial or complete training of many neural network architectures in the search space. Such methods are infeasible for very large architecture design spaces, as considerable sampling and training of neural networks would be required to sufficiently describe the entire design space. Popular NAS techniques often involve generating and training neural network architectures and using this accuracy to train accuracy predictors to serve as feedback in the search algorithm. This architecture generation and feedback methodology can be leveraged by algorithms such as reinforcement learning (Luo et al. [2019]) and evolutionary algorithms (Real et al. [2019]). One-shot methods (Cai et al. [2019, 2020], Xie et al. [2020], Yang et al. [2020], Liu et al. [2019]) maintain a super-network, from which sub-networks are sampled. The weights are shared between the sub-networks which reduces NAS cost as there are lesser parameters to train.

36th Conference on Neural Information Processing Systems (NeurIPS 2022).

The primary challenge of Neural Architecture Search is the evaluation of candidate architectures, which can take hours to days to estimate (Abdelfattah et al. [2021]). Further, practical deployment of neural networks is no longer limited to accuracy-oriented neural architecture design. The need for efficient deployment has given rise to significant literature in co-design of hardware and neural network architectures (Choi et al. [2021], Akhauri et al. [2021], Zhang et al. [2020]). However, hardware performance metrics in such co-design tasks are often non-differentiable in nature due to factors such as memory access patterns and cache hierarchy. The interaction between the workload and the hardware makes this optimization problem significantly more complex, and being able to score neural network architectures without the expensive training step would allow for the development of more efficient co-design algorithms.

Most zero-cost proxies utilize a single minibatch of data and a single forward/backward propagation pass to score a neural network (Abdelfattah et al. [2021]). We refer to algorithms that score neural network architectures without training as *Zero-Cost Neural Architecture Scoring Metrics* (ZC-NASM). Design of existing ZC-NASMs are driven by human intuition or are theoretically inspired. These algorithms attempt to quantify the *trainability* and *expressivity* of neural networks. The primary hurdle with existing ZC-NASM design is that they are not generalizable to different neural architecture design spaces.

In this paper, we introduce a genetic programming driven methodology to automatically discover ZC-NASMs that are interpretable, generalizable and deliver state of the art score-accuracy correlation. We refer to our method as EZNAS (**E**volutionarily Generated **Z**ero-Cost **N**eural **A**rchitecture **S**coring Metric). Our approach is:

- *Interpretable:* We discover ZC-NASMs as expression trees that explicitly indicate which NN features and mathematical operations are being utilized.

- *Generalizable:* The ZC-NASM discovered by EZNAS delivers **SoTA score-accuracy correlation on unseen neural architecture design spaces and datasets (NASBench-201, NDS & NATS-Bench).** Our framework utilizes simple mathematical operations and an expression tree structure that can be trivially extended to implement arbitrarily complex ZC-NASMs

- *Efficient:* We are able to discover our SoTA ZC-NASM on an Intel(R) Xeon(R) Gold 6242 CPU in under 24 hours. This requires **12.5× lesser CO2e than a NAS search.** Our ZC-NASM is generalizable to multiple design spaces, increasing the end-to-end efficiency of NN architecture search by over two orders of magnitude.

## 2 Related Work

**Neural Architecture Search (NAS)** was proposed to reduce the human effort that goes into manually designing complex neural network architectures. Early efforts in the field of NAS adopted brute force methods by training candidate architectures and using the obtained accuracy as a proxy to discover better architectures. AmoebaNet (Real et al. [2019]) utilized evolutionary algorithms (EA) and 3150 GPU days of compute to achieve 74.5% top-1 accuracy on the ImageNet dataset. Many EA and Reinforcement Learning (RL) driven methods have since significantly improved the efficiency of the search process (Yang et al. [2020], Liu et al. [2018], Tan et al. [2019], Zoph et al. [2018]). These methods often require sampling and training of several candidate architectures. One-shot (Hu et al. [2020], Xie et al. [2020], Cai et al. [2019]) methods of NAS do not require training of candidate architectures to completion, but train large super-networks and identify sub-networks that can give high accuracy. Such super-networks can be generated automatically from pre-trained models (Muñoz et al. [2021]). These improvements have significantly reduced the cost of NAS. However, as the search space continues to get larger with new architectural innovations, we need more efficient methods to predict the accuracy of neural networks in intractably large design spaces.

**Zero-Cost Neural Architecture Scoring** is a promising paradigm which explores zero-cost proxies for estimating the true accuracy of neural networks. The majority of research in this field introduces novel methods of scoring neural networks at initialization along with a theoretical or intuitive explanation of their scoring strategy. For instance, (Abdelfattah et al. [2021]) empirically studies metrics such as `synflow`, `SNIP` and `FISHER`. Figure 2 depicts the expression tree representations of these metrics. NASWOT (Mellor et al. [2021]) works by treating the output of each layer as a binary

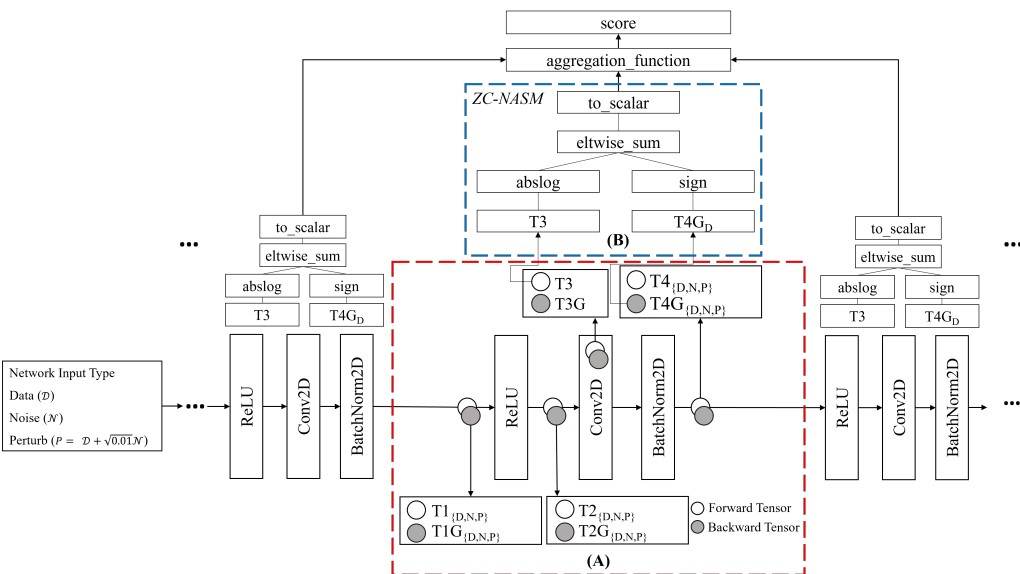

Figure 1: **(A)**: Collecting neural network statistics for the ZC-NASM. The $\mathcal{D}$ input tensor represents a single mini-batch from the dataset serving as input to the neural network. $\mathcal{N}$ represents a randomly initialized noise tensor. $\mathcal{P}$ represents an input to the neural network which is a data-sample perturbed by noise. NN statistics are collected for each of the $\mathcal{D}$, $\mathcal{N}$ and $\mathcal{P}$ inputs. **(B)**: The ZC-NASM is applied to each `ReLU-Conv2D-BatchNorm2D` (RCB) instance of the neural network. The ZC-NASM has 22 tensors available to it as arguments in each RCB instance, generated by collecting intermediate tensors of the neural network for the three types of input ($\mathcal{D}, \mathcal{N}, \mathcal{P}$). The ZC-NASM depicted above only utilizes 2 intermediate tensors ($T3$ and $T4G_P$) in each layer to generate a score.

indicator (zero if value is negative, one if value is positive) and using the hamming distance between two binary codes induced by an untrained network from two inputs as a measure of dissimilarity. The intuition here is that the more similar two inputs are, the more challenging it is for the network to learn how to separate them. Works like TENAS (Chen et al. [2021]), GradSign (Zhang and Jia [2021]) create proxies for *trainability* & *expressivity* of neural networks to rank them.

**Program Synthesis**: Program synthesis is the task of automatically discovering programs that satisfy user constraints. AutoML-Zero (Real et al. [2020]) evolves entire machine learning algorithms from scratch with little restrictions on form and using only simple mathematical operations. The algorithms are learnt symbolically, representing programs as a sequence of instructions. We posit that the current endeavor of identifying fundamental architectural properties that correlate strongly with testing accuracy can benefit from a methodology that minimizes human bias and intervention in a similar fashion. This stems from the observation that majority of the existing ZC-NASMs can be represented as simple programs that utilize the neural network statistics as inputs which can be discovered with genetic programming.

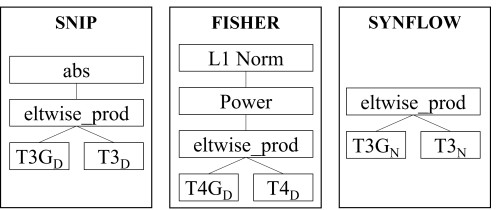

Figure 2: Expression tree representation of existing ZC-NASMs. $T3G_D$ represents the Conv2D weight gradient for a mini-batch of data, $T3_D$ represents the weights of Conv2D. $T4G_D$ represents the activation gradient and $T4_D$ represents the corresponding activation. $T3G_N$ represents the Conv2D weight gradient for a mini-batch of random noisy tensor, $T3_N$ represents the weights of Conv2D. Each program is applied to all layers of the neural network, and the output is aggregated to generate the final score. Further details of this program representation is explained later.

## 3 Evolutionary Framework

EZNAS uses evolutionary search to discover programs that can score neural network architectures such that it correlates highly with accuracy. In this section, we detail the specifics of how ZC-NASM programs are constructed, evaluated and evolved with EZNAS.

## 3.1 Program Representation

Each individual ZC-NASM has to be evaluated on tens of gigabytes of tensors of NN architectures with no approximations to generate precise ZC-NASM score-accuracy correlation to serve as the measure of fitness. To make such search computationally tractable, it is crucial to not introduce redundant operations in the program. Our initial attempt described in the appendix resembled that of AutoML-Zero (Real et al. [2020]) where ZC-NASMs were represented as a sequence of instructions and a memory space to store intermediate tensors. This led to discovery of ZC-NASMs with many redundant computations due to program length bloating and an intractably large run-

---

**Algorithm 1** EZNAS Search Algorithm

1: evol_space = [$NN_{1,enas}$, $NN_{2,enas}$,... $NN_{N,enas}$]
2: population = gen_random_valid_population(n)
3: # Evaluate and assign fitness over evol_space.
4: population = evaluate(population)
5: **for** gen=1:T **do**
6:     offspring = []
7:     **while** len(offspring) $< \lfloor n/2 \rfloor$ **do**
8:         # Variation Function
9:         children = VarOr(population)
10:         offspring.append(selectValid(children))
11:     new_c = gen_random_valid_population($\lfloor n/2 \rfloor$)
12:     offspring.append(new_c)
13:     population = evaluate(offspring)

---

time. To reduce the complexity of search, we necessitate a *expression tree* structure on the ZC-NASM program to capture the executional ordering of the program. The program output appears at a root node, and the child (terminal) nodes are the *arguments* of the expression tree. These *arguments* are the *network statistics*. The advantage of this program representation is that there is only a single root node with dense connectivity from the root to the terminal nodes, leading to fewer redundancies.

## 3.2 Neural Network Statistics Generation

Each ZC-NASM requires a set of *arguments* as inputs. These arguments are the intermediate tensors of the neural network. As depicted in Figure 1, for any sampled neural network from the NDS or NASBench-201 spaces, we identify every `ReLU-Conv2D-BatchNorm2D` (RCB) instance at initialization, and register the activations, weights and the corresponding gradients for three types of network inputs (a mini-batch of data ($\mathcal{D}$), a random noisy tensor ($\mathcal{N}$), and a mini-batch of data perturbed by random noise ($\mathcal{P}$)).

We collect $T1_{\{D,N,P\}}$, $T2_{\{D,N,P\}}$, $T3$, $T4_{\{D,N,P\}}$ (10 tensors) and $T1G_{\{D,N,P\}}$, $T2G_{\{D,N,P\}}$, $T3G_{\{D,N,P\}}$, $T4G_{\{D,N,P\}}$ (12 tensors) from each RCB instance. T3 represents the Conv2D weights and does not change for the network input type, thus giving us 22 tensors for each RCB instance. We present an alternate formulation by identifying `Conv2D-BatchNorm2D-ReLU` (CBR) instance for network statistics generation to demonstrate that the evolutionary framework is not restricted to a RCB structure in the appendix.

## 3.3 Mathematical Operations

The expression tree describes the execution order of the mathematical operations available to us. It is crucial to provide a varied set of operations to process the neural network statistics effectively. We provide 34 unique operations in our program search space. We include basic mathematical operations (Addition, Product) as well as some advanced operations such as Cosine Similarity and Hamming Distance. We provide the full list of mathematical operations in the supplementary material.

## 3.4 Program Application

Majority of the neural network architectures available in NASBench-201 and NDS have over 100 instances of `ReLU-Conv2D-BatchNorm2D`. This may mean that an expression tree would have as many as 2200 ($22 \times 100$) possible arguments (terminal nodes), each of which can be used multiple times. This would result in a computationally intractable expression tree. To simplify the search problem, we generate a single expression tree with 22 possible inputs. It is not necessary that the root node of an expression tree would give a scalar value, so we add a `to_scalar` operation above the root node of the expression tree. As seen in Figure 1 the expression tree is then applied on every `ReLU-Conv2D-BatchNorm2D` instance and the output is aggregated across all instances using an `aggregation_function`. This serves as the 'score' of the the sampled neural network architecture. In EZNAS-A, the `aggregation_function` and `to_scalar` are both `mean`. In the appendix, we explore L2-Norm as a `to_scalar` function as well and find that we are able to discover effective ZC-NASMs.

### 3.5 Evolutionary Algorithm

Our search algorithm discovers programs by modifying the expression tree representation of the population by `variation_functions`. A fitness score is generated for every expression tree in the population at each generation. The `variation_function` (VarOr) is then used to generate new offspring. We describe our search algorithm in Algorithm 1

#### 3.5.1 Population Initialization

We initialize a population of `n` programs. We do not impose any restrictions on the operations the nodes can use in the expression tree. Due to this, the number of valid expression trees is several orders of magnitude lesser than the total number of expression trees that can be generated with our

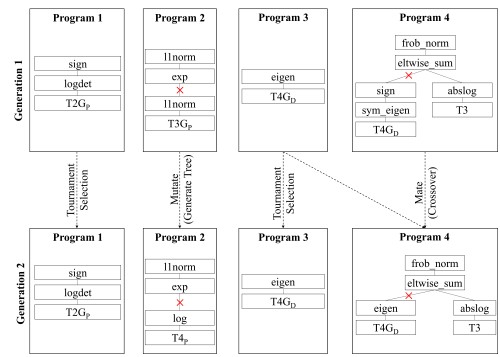

Figure 3: Variation of expression trees every generation. The x signs denote point of cross-over or mutation.

mathematical operators and network statistics. To increase search efficiency, we would like to ensure that we sample and evolve only valid expression trees. To enable this, we ensure that all individuals in the population are valid programs by testing program execution on a small sub-set of network statistics data. All programs that produce outputs with $inf$, $nan$ or fail to execute are replaced by new randomly initialized valid programs.

#### 3.5.2 Fitness Objective

As the goal of zero-cost NAS is to be able to rank neural network architectures well, we utilize the Kendall Tau rank correlation coefficient as the fitness objective. The search objective is to maximize the Kendall Tau rank correlation coefficient between the scores generated by a program and the test accuracy of the neural networks in the `evol_space`.

#### 3.5.3 Variation Algorithms

In our tests, we utilize the VarOr implementation from *Distributed Evolutionary Algorithms in Python* (DEAP) (Fortin et al. [2012]) framework for the variation of individual programs. We generate n (hyper-parameter) offspring programs at each generation. $\lfloor n/2 \rfloor$ offspring are generated as a result of three operations; crossover, mutation or reproduction. These variations are depicted in Figure 3. For crossover, two individual programs are randomly selected from the population and mated. Our mating function randomly selects a crossover point from each individual and exchanges the sub-trees with the selected point as root between each individual. The first child is appended to the offspring population. For mutation, we randomly select a point in the individual program, and replace the sub-tree at that point by a randomly generated expression tree. We repeat the variation algorithm on the population till $\lfloor n/2 \rfloor$ valid individuals are generated. To encourage diversity, we also randomly generate $\lfloor n/2 \rfloor$ valid individuals. We have placed static limits on the depth of all expression trees at 10.

### 3.6 Program Evaluation Methodology

At each generation, the fitness of the entire population is invalidated and recalculated. Calculating the fitness of each program on the entire dataset (which can be approximately 1 TB) is computationally infeasible. Further, we may want to find generalized programs that give high fitness on many different architecture design spaces and datasets. This would translate to evaluating each individual on over 7 TB of data on the NDS and NASBench-201 search spaces.

Reducing the computation by evaluating the fitness of the population on a single small fixed sub-set of neural networks from the search space causes discovered programs to trivially over-fit to the sub-set statistics in our tests. To address over-fitting of programs to small datasets of network statistics while minimizing compute resources required for evaluating on the entire dataset of network statistics, we generate an *evolution task dataset*. This is generated by randomly sampling 80 neural networks

| SPEARMAN $\rho$ | CF10 | CF100 | IN16-120 |
|---|---|---|---|
| EZNAS-A | **0.83** | **0.82** | **0.78** |
| synflow | 0.74 | 0.76 | 0.75 |
| jacob_cov | 0.73 | 0.71 | 0.71 |
| FLOPs | 0.75 | 0.72 | 0.69 |
| Params | 0.75 | 0.72 | 0.69 |
| grad_norm | 0.58 | 0.64 | 0.58 |
| snip | 0.58 | 0.63 | 0.58 |
| grasp | 0.48 | 0.54 | 0.56 |
| fisher | 0.36 | 0.39 | 0.33 |

Table 1: Spearman $\rho$ for NASBench-201.

| SPEARMAN $\rho$ | CF10 | CF100 | IN16-120 |
|---|---|---|---|
| EZNAS-A | **0.89** | **0.74** | **0.81** |
| NASWOT | 0.45 | 0.18 | 0.41 |

Table 2: Spearman $\rho$ for NATS-Bench-SSS.

| KENDALL $\tau$ | CF10 | CF100 | IN16-120 |
|---|---|---|---|
| EZNAS-A | **0.65** | **0.65** | **0.61** |
| NASWOT | 0.57 | 0.61 | 0.55 |
| AngleNAS | 0.57 | 0.60 | 0.54 |
| FLOPs | 0.56 | 0.54 | 0.50 |
| Params | 0.56 | 0.54 | 0.50 |

Table 3: Kendall $\tau$ for NASBench-201.

| KENDALL $\tau$ | DARTS | Amoeba | ENAS | PNAS | NASNet |
|---|---|---|---|---|---|
| EZNAS-A | **0.56** | **0.45** | **0.52** | **0.51** | **0.44** |
| NASWOT | 0.47 | 0.22 | 0.37 | 0.38 | 0.30 |
| grad_norm | 0.28 | -0.1 | -0.02 | -0.01 | -0.08 |
| synflow | 0.37 | -0.06 | 0.02 | 0.03 | -0.03 |
| FLOPs | 0.51 | 0.26 | 0.47 | 0.34 | 0.20 |
| Params | 0.50 | 0.26 | 0.47 | 0.32 | 0.21 |

Table 4: Kendall $\tau$ for NDS CIFAR-10.

from each available search space (NASBench-201 and NDS) and dataset (CIFAR-10, CIFAR-100, ImageNet-16-120). We evaluate individuals by randomly choosing s search spaces, and sampling 20 neural networks from each of the chosen search spaces. We take the minimum fitness achieved by the individual program on the s spaces. We consider s as a hyper-parameter. In our tests, this is consistently kept at 4.

### 3.7 Program Testing Methodology

At the end of the evolutionary search, our primary goal is to test whether the programs discovered are able to provide high fitness on previously unseen neural network architectures. We test the fittest program from our final population as well as the two fittest programs encountered through-out the evolutionary search. At test time, we take the program and find the score-accuracy correlation over 4000 neural network architectures sampled from the NASBench-201 and NDS design spaces.

## 4 Results

As described in Algorithm 1, we search for effective ZC-NASMs on every design space from NDS CIFAR-10 and the NASBench-201 datasets. Each search takes approximately 24 hours on a Intel(R) Xeon(R) Gold 6242 CPU with 1 terabyte of RAM.

We choose the most consistent ZC-NASM (EZNAS-A, discovered by evolving programs on the NDS-DARTS CIFAR-10 search space) from our evolutionary search and compare it with existing ZC-NASMs from recent works. We report both the Kendall $\tau$ rank correlation coefficient and the Spearman $\rho$ rank-order correlation coefficient to fairly compare EZNAS-A with a broad set of ZC-NASMs from recent literature. We provide our Spearman $\rho$ correlation on the NATS-Bench SSS search space and NDS ImageNet design spaces as well.

**NASBench-201:** We report our score-accuracy correlation by scoring all neural

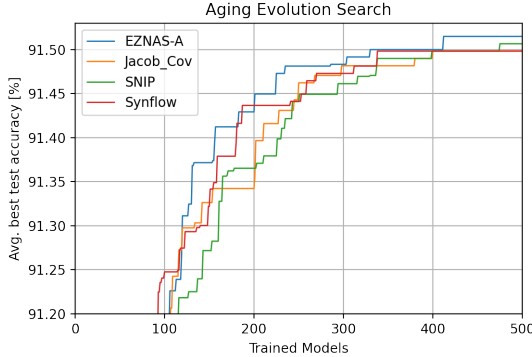

Figure 4: Search speedup with EZNAS-A on CIFAR-10 NAS-Bench-201. The average best test accuracy is taken over 10 repeated experiments.

networks in the NASBench-201 design space. As seen in Table 3, we obtain the highest score-accuracy Kendall Tau Ranking Correlation on all datasets. We also benchmark our work against the pruning based saliency metrics formalized in (Abdelfattah et al. [2021]) and find that our program EZNAS-A has the highest Spearman $\rho$ among the ZC-NASMs in Table 1.

**NDS:** We report our score-accuracy correlation by scoring all neural networks in the NDS design space for the CIFAR-10 dataset and scoring 40 random neural networks over 5 seeds for the ImageNet dataset (Table 5). We find that EZNAS-A has the highest score accuracy Kendall Tau Ranking Correlation on ea ch of the design spaces of NDS CIFAR-10 in Table 4.

**NATS-Bench:** We report our score-accuracy correlation by scoring 200 randomly sampled neural networks in the NATS-Bench TSS (same as NB-201) and SSS (Dong et al. [2021]) design space averaged over 5 seeds for the CIFAR-10, CIFAR-100 and ImageNet16-120 dataset in Table 2.

**Neural Architecture Search Integration:** We integrate our ZC-NASMs EZNAS-A with the Aging Evolution (AE) Search algorithm from (Real et al. [2019]). Our evolutionary search discovers a ZC-NASM that is competitive in search efficiency with existing ZC-NASMs in Figure 4.

## 5  Examination Of Best Programs

EZNAS discovers a set of ZC-NASMs at the end of the evolutionary search. In this section, we analyze two of the best ZC-NASMs discovered by our method to give a deeper understanding of the nature of programs. We

| SPEARMAN $\rho$ | DARTS | Amoeba | PNAS | ENAS | NASNet |
|---|---|---|---|---|---|
| EZNAS-A | **0.70** | **0.58** | **0.43** | **0.43** | **0.31** |

Figure 5: Spearman $\rho$ for NDS ImageNet.

refer to these programs as `EZNAS-A` and `EZNAS-B`. For each program, we generate random input tensors of varying sizes and average the ZC-NASM score 5000 times. This is an approximate method to understand how the ZC-NASM responds to change in architectural parameters (kernel size, number of channels, activation size).

`EZNAS-A:` The evolution task dataset for discovering this program was NDS-DARTS. The input to the ZC-NASM in Figure 6 is the $T3G_N$ (Weight Gradient with Random Noise Input) tensor. We find that the score increases as the number of channels or depth increases, we also observe that kernel sizes of 1 and 7 give higher scores than 3 and 5 with the lowest score being assigned to kernel of size 3. It is interesting to see that the expectation value of `EZNAS-A` in Figure 6 translates to a weighted form of parameter counting, with a non-linear monotonically increasing scaling of score with the number of input/output channels and a locally parabolic relationship between the score and the kernel size with the minimum score at kernel size of 3.

`EZNAS-B:` The evolution task dataset for discovering this program was NDS-ENAS. The input to the ZC-NASM in Figure 6 is $T1G_P$ (Difference in pre-activation gradients for a noise perturbed data mini-batch). We find that the activation map size is exponentially more influential to the score when compared to the number of channels.

Through this analysis, we see that `EZNAS-A` & `EZNAS-B` generate a score which is correlated with the activation or weight sizes. It is interesting to note that when compared to ZC-NASMs from recent literature, this form of weighted parameter counting works more consistently. This is supported by our finding that FLOPs and Params are more generalizable than ZC-NASMs which work on the NASBench-201 space but fail on the NDS space.

## 6  Discussion and Future Work

With the EZNAS formulation, we discover effective ZC-NASMs which generalize across design spaces and datasets, as well as give SoTA score-accuracy correlation. This is a promising new direction, but there is much work to be done. In this section, we discuss the current limitations of EZNAS with scope for future work.

**Program Design:** To simplify the search problem, we take a mean (`aggregation_function`) of the scores generated across all `ReLU-Conv2D-BatchNorm2D` instances. This formulation does not take layer connectivity patterns into account explicitly. Extending our evolutionary

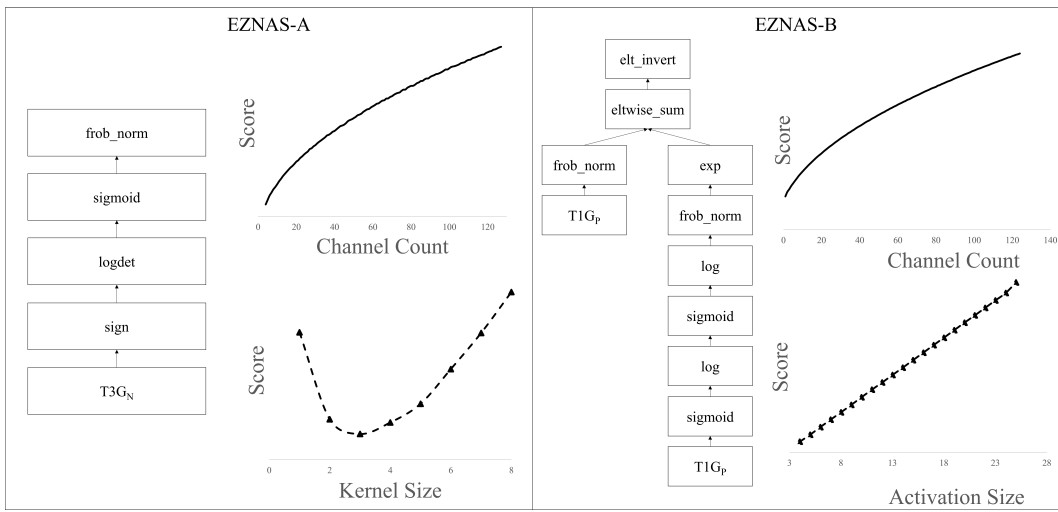

Figure 6: (Left) Analysis of our best program (`EZNAS-A`) on the Image Classification task. `to_scalar` is not required as the output is already a scalar. (Right) Analysis of our second best program (`EZNAS-B`) on the Image Classification task.

search to take into account the global connectivity pattern of the neural networks or learning weighted averaging techniques for the individual layer (RCB instance) scores could help us discover better ZC-NASMs. Further, we impose structural restrictions on our program by necessitating a fixed structure (`ReLU-Conv2D-BatchNorm2D`). We have to truncate instances of `ReLU-Conv2D-Conv2D-BatchNorm2D` from the NDS space by ignoring the second convolution. As more diverse architectures are introduced in the field, we must re-formulate collection of network statistics in a more robust manner. Note that NASWOT measures the similarity in the binary codes generated from the ReLU units for two different inputs. Our program design limits us to only using a single mini-batch of inputs and does not compare inputs directly. Thus, we would not be able to discover the NASWOT ZC-NASM metric with our approach. Fortunately, integrating such functionality into the program design space is feasible and is an interesting line of future work to generate more complex metrics.

It is also important to have a robust and diverse set of mathematical operations to discover effective ZC-NASMs. We utilize 34 mathematical operations inspired by AutoML-Zero (Real et al. [2020]) and operations found in existing ZC-NASMs. None of our operations or network statistics generation have any scalar hyper parameters. For example, the `Power` operation is fixed to do an element-wise square, and the noise generation for input tensor is fixed to $\mathcal{N}(0, 1)$. Optimizing these values dynamically and ensuring a sufficiently diverse set of operations may enable discovery of better ZC-NASMs.

**Network Statistics:** Due to the computational resources required to generate network statistics, we have to pre-compute them and use the generated data as an *evolutionary task dataset*. We only use network statistics with input batch-size of 1 for evolution. A single sample statistic of a neural network is insufficient to describe the architecture, as evidenced by our increase in score-accuracy correlation with the batch size in Figure 7. Further, from Figure 7 we observe high variance in the fitness of a ZC-NASM when evaluated for multiple seeds with a batch size of 1. Testing over a large number of seeds or increasing the batch-size would cause a linear increase in memory requirement as well as increase in run-time of the evolutionary search.

**Ranking Architectures:** It is interesting to observe that in the entire neural architecture design space, FLOPs and Params are competitive proxies and sometimes better than many existing ZC-NASMs. A deeper investigation reveals that ranking the top 10% of the neural networks in the design space is a significantly more difficult task. In the top 10% of neural networks, FLOPs and Params are extremely weak proxies. Further, (Abdelfattah et al. [2021]) finds that among their ZC-NASMs, `synflow` is the only one which is able to serve as a weak correlator for performance in top 10% of the neural networks on the NASBench-201 CIFAR-100 and ImageNet16-120 design spaces. However, `synflow` is not able to rank networks effective in the top 10% of CIFAR-10 NASBench-201 and the entire NDS design space.

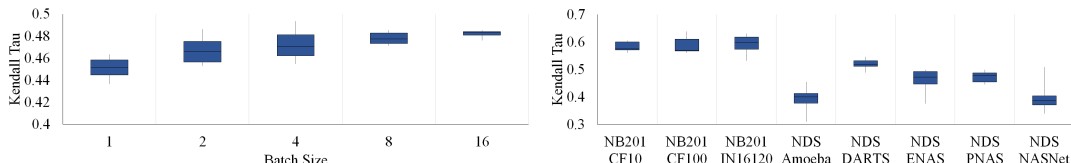

Figure 7: (Left) Effect on score-accuracy correlation of `EZNAS-A` with respect to batch size. This test was done on the NDS PNAS design space over 7 seeds for 400 Neural Networks.(Right) Effect of seed on our score-accuracy correlation of `EZNAS-A` with a batch size of 1. CIFAR and ImageNet abbreviated as CF and IN respectively. This test was done on each design space over 7 seeds for 400 Neural Networks.

`EZNAS-A` is able to rank neural networks robustly across both the NASBench-201 and NDS design spaces, but does not rank the top 10% of neural networks effectively. We discover alternative ZC-NASMs that can weakly rank the top 10% of neural networks, but such ZC-NASMs fail to generalize across neural architecture design search spaces. This ability to identify good architectures but inability to distinguish between the best architectures may be an inherent limitation of ZC-NASMs. Alternatively, improvements to our evolutionary framework may result in robust ZC-NASMs that can rank the top 10% neural networks effectively. This is an interesting research direction and may require creation of more neural architecture design spaces like NASBench-201/NDS which exhibit lower correlation with FLOPs/Params.

**Evolutionary Search:** In this paper, we introduce an intuitive program representation and appropriate variation methodologies to enable discovery of ZC-NASMs with minimal human intervention. However, recent advances in the field of Neuro-Symbolic Program Synthesis (Parisotto et al. [2017]) to learn mappings from input-output examples (neural network statistics to neural architecture scores) can motivate improvements in our evolutionary search. Further, exploring architecture connectivity encodings (White et al. [2021]) for neural architecture search and discovering programs to rank neural architecture encodings may enable discovery of effective connectivity patterns to enable a deeper understanding of important features in neural architecture design beyond those contained in activations maps and weights.

**Efficiency:** EZNAS is able to discover a set of ZC-NASMs on a single CPU in under 24 hours, which translates to 358.6 g of $CO_2e$ (Lannelongue et al. [2020]). In comparison, a single NAS search (assuming 8 GPU days (Zhou et al. [2020])) can take over 4.49 kg of $CO_2e$. Recent works integrate their zero-cost proxies with differentiable architecture search (Xiang et al. [2021]) to deliver $40\times$ NAS speed-up. As the architecture design spaces that ZC-NASMs need to be discovered on diversify, the efficiency of our methodology must be improved to scale to larger problems. Seeding the initial population with viable candidates can enable faster discovery of robust ZC-NASMs. The search process can be made more efficient by using lower precision numerical formats, or exploring proxy tasks on smaller datasets (down-sampled image datasets, smaller neural network design architecture etc.).

## 7    Conclusion

In this paper, we present EZNAS, a novel genetic programming driven approach to discover *Zero-Cost Neural Architecture Scoring Metrics (ZC-NASMs)*. The key advantage of EZNAS is that it is an interpretable approach to discover generalizable methods to rank neural networks. *Generalizability* of a ZC-NASM is crucial for its practical utility, as a robust ZC-NASM should be able to rank NNs in previously unseen neural architecture search spaces.

With our approach, we are able to discover a ZC-NASM (`EZNAS-A`) which evolved only on the NDS DARTS space, but delivers **state of the art score-accuracy correlation across both the NASBench-201 and NDS design space.** We also demonstrate the generalizability of `EZNAS-A` by providing extremely strong correlation results on NATS-Bench as well as NDS ImageNet. We demonstrate competitive search efficiency of `EZNAS-A` by integrating our metric with the Aging Evolution (AE) search algorithm from (Abdelfattah et al. [2021]). Further, we provide an in-depth analysis of the nature of programs we have discovered, along with a detailed account of existing limitations of our methodology to motivate future work in this direction. EZNAS has the potential to reduce human bias in design of ZC-NASMs, and aid in discovery of programs that provide deeper insights into properties that make a neural network perform well.

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
