# EZNAS: Evolving Zero-Cost Proxies For Neural Architecture Scoring

**Yash Akhauri**[1]   **J. Pablo Muñoz**[2]   **Nilesh Jain**[2]   **Ravi Iyer**[2]
[1]Cornell University   [2]Intel Labs
ya255@cornell.edu
{pablo.munoz,nilesh.jain,ravishanker.iyer}@intel.com

## 1   Appendix

### 1.1   NASBench-201 and NDS

For image classification, we utilize the NASBench-201 Dong and Yang [2020] and NDS Radosavovic et al. [2019] NAS search spaces for our evolutionary search as well as testing. NASBench-201 consists of 15,625 neural networks trained on the CIFAR-10, CIFAR-100 and ImageNet-16-120 datasets. Neural Networks in Network Design Spaces (NDS) uses the DARTS Liu et al. [2019] skeleton. The networks are comprised of cells sampled from each of AmoebaNet Real et al. [2019], DARTS Liu et al. [2019], ENAS Pham et al. [2018], NASNet Zoph et al. [2018] and PNAS Liu et al. [2018]. There exists approximately 5000 neural network architectures in each NDS design space.

### 1.2   Sequential Program Representation

Our initial attempts at discovering ZC-NASMs took a different approach to program representation. The *sequential program representation* described in Figure 1 posed no structural limitations on the program. We have 22 static memory addresses, which contained network statistics and are referenced with integers 0-21. To store intermediate tensors generated by the program, we allocate 80 dynamic memory addresses, which can be over-written during the program execution as well. To store intermediate scalars generated by the program, we allocate 20 memory addresses. As seen in Figure 1, we represent the programs as integers, where each instruction is expressed as 4 integers. The first integer provides the write address, the second integer provides the operation

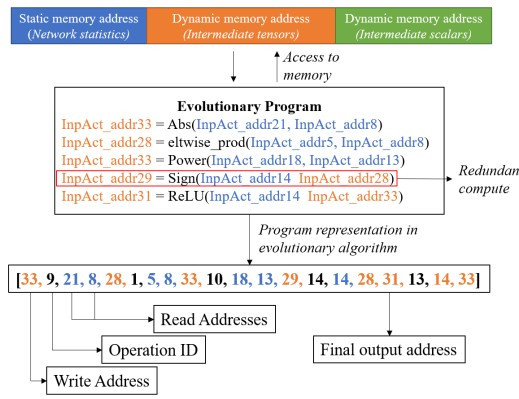

Figure 1: Our sequential program representation.

ID and the third and fourth integers provide the read addresses for the operation. We initialize *valid* random integer arrays and convert them to programs to evaluate and fetch the fitness (score-accuracy correlation). We allow `Mate`, `InsertOP`, `RemoveOP`, `MutateOP` & `MutateOpArg` as variation functions. The `Mate` function takes two individuals, and takes the first half of each individual. Then, these components are interpolated to generate a new individual. The `InsertOP` function inserts an operation at a random point in the program. The `RemoveOP` function removes an operation at a random point in the program. The `MutateOP` changes a random operation in program without changing read/write addresses. The `MutateOpArg` function simply replaces one of the read arguments of any random instruction with another argument from the same address space (dynamic address argument cannot be replaced by a scalar address argument).

36th Conference on Neural Information Processing Systems (NeurIPS 2022).

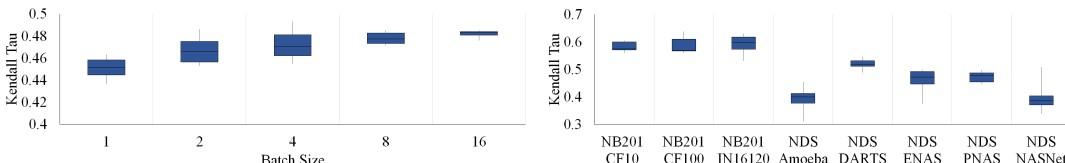

Figure 2: (Left) Experiment demonstrating the effect of seed on our score-accuracy correlation for neural network initialization with a batch size of 1. (Right) Effect of seed on our score-accuracy correlation with a batch size of 1. CIFAR and ImageNet abbreviated as CF and IN respectively. This test was done on each design space over 7 seeds for 400 Neural Networks.

While we are able to discover weak ZC-NASMs with this formulation, we observe that there are too many redundancies in the programs discovered. Program length bloating as well as operations that do not contribute to the final output were frequently observed. Due to these issues, the evolution time evaluation of individual fitness quickly became an intractable problem. To address this, we change our program representation to a expression tree representation in the results reported in the paper. This representation necessitates contribution of each operation to the final output, which means there is no redundant compute. While the sequential program representation is valid, we believe that significant engineering efforts are required to ensure discovery of meaningful programs. Our sequential program representation is directly inspired by the formulation used in AutoML-Zero Real et al. [2020]. AutoML-Zero makes significant approximations in the learning task to evolutionarily discover MLPs. While AutoML-Zero has a much larger program space to search for, approximations in computing individual fitness are not feasible in our formulation as generating exact score-accuracy correlation is an important factor in selecting individuals with high fitness.

### 1.3 Noise and Perturbation for Network Statistics

To generate network statistics, we use three types of input data. The first is simply a single random sample from the dataset (e.g. a single image or a batch of images from CIFAR-10). To generate a noisy input, we simply use the default `torch.randn` function as `input = torch.randn(data_sample.shape)`. The third type of input we provide is a data-sample which has been perturbed by random noise (`input = data_sample + 0.01**0.5*torch.randn(data_sample.shape)`).

### 1.4 Network Initialization Seed Test

In Figure 2 (Left), we use different seeds to change the initialization and input tensors, but keep the neural architectures being sampled fixed in the respective spaces. The variance in the score accuracy correlation is much lesser than in Figure 2 (Right) where the seed also controls the neural architectures being sampled. This shows the true variation in our `EZNAS-A` ZC-NASM with respect to network initialization.

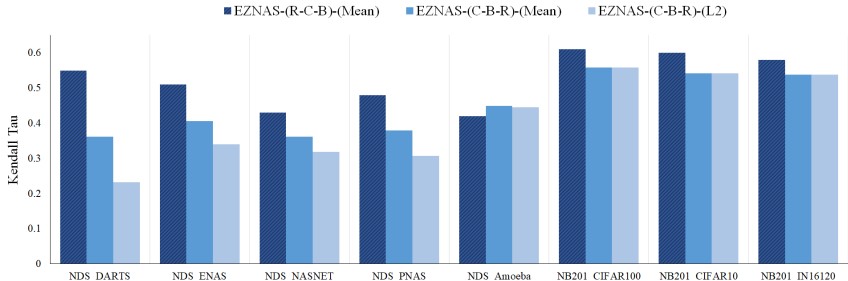

Figure 3: Experiment demonstrating the ability to discover ZC-NASMs with an alternate network statistics collection strategy and `to_scalar` function. Experiments are named as EZNAS-(Statistics Collection Structure)-(to_scalar). Two statistics collection structures are tested. (R-C-B) is a `ReLU-Conv2D-BatchNorm2D` structure, (C-B-R) is a `Conv2D-BatchNorm2D-ReLU` structure. (`to_scalar`) can be Mean or L2.

| Kendall Tau | NASBench201 CIFAR10 | NASBench201 CIFAR100 | NASBench201 ImageNet16-120 | Amoeba CIFAR10 | DARTS CIFAR10 | ENAS CIFAR10 | PNAS CIFAR10 | NASNet CIFAR10 | FLOPs | Params |
|---|---|---|---|---|---|---|---|---|---|---|
| NASBench201 CIFAR10 | 0.533 | -0.499 | 0.4345 | 0.4422 | **0.5777** | 0.472 | 0.4514 | 0.2633 | _0.5610_ | 0.5610 |
| NASBench201 CIFAR100 | 0.5242 | -0.496 | 0.4522 | 0.4152 | **0.5627** | 0.476 | 0.453 | 0.2531 | _0.5472_ | 0.5472 |
| NASBench201 ImageNet16-120 | 0.4575 | -0.424 | 0.4350 | 0.4213 | **0.5733** | 0.456 | 0.2903 | 0.2338 | _0.5036_ | 0.5036 |
| Amoeba CIFAR10 | 0.1146 | -0.029 | 0.3263 | 0.352 | _0.3774_ | 0.380 | 0.35278 | **0.4042** | 0.2614 | 0.2658 |
| DARTS CIFAR10 | 0.3236 | -0.173 | 0.2088 | 0.4187 | _0.5077_ | 0.460 | 0.47570 | 0.1439 | **0.5079** | 0.5042 |
| ENAS CIFAR10 | 0.2833 | -0.143 | 0.3132 | 0.4292 | 0.3795 | 0.422 | 0.41701 | 0.3400 | **0.4739** | _0.4704_ |
| PNAS CIFAR10 | 0.2208 | -0.053 | 0.4212 | 0.4466 | 0.4435 | **0.514** | _0.4874_ | 0.3799 | 0.3363 | 0.3223 |
| NASNet CIFAR10 | 0.2370 | 0.0229 | 0.2880 | _0.3789_ | **0.4381** | 0.364 | 0.3594 | 0.3757 | 0.1996 | 0.2102 |

Figure 4: Full correlation table. Each column represents the dataset evolution was performed on. The DARTS-CIFAR10 column is the `EZNAS-A` NASM. Each row represents the dataset the best discovered NASM program was tested on. Best score-accuracy KTR in bold and underlined. Second best score-accuracy KTR in italics and underlined. These tests are done by evolving on 100 neural networks and testing on the test task dataset (1000 randomly sampled neural networks on NASBench-201 and 200 randomly sampled neural networks on NDS). The network statistics were generated with a batch size of 1.

## 1.5 Hardware used for evolution and testing

Our evolutionary algorithm runs on Intel(R) Xeon(R) Gold 6242 CPU with 630GB of RAM. Our RAM utilization for evolving programs on a single Image Classification dataset was approximately 60GB. RAM utilization can vastly vary (linearly) based on the number of neural network statistics that are being used for the evolutionary search. Our testing to generate the statistics for the seed experiments as well as the final Spearman $\rho$ and Kendall Tau Rank Correlation Coefficient is done on an NVIDIA DGX-2 server with 4 NVIDIA V-100 32GB GPUs.

## 1.6 Varying network statistics and `to_scalar`

In Figure 3, we detail 3 tests while evolving on the NDS_DARTS CIFAR-10 search space in an identical fashion to `EZNAS-A` (referred to as `EZNAS-(R-C-B)-(Mean)` here). `EZNAS-(C-B-R)-(Mean)` and `EZNAS-(C-B-R)-(L2)` correspond to alternate `to_scalar` and network statistics collection tests respectively. We demonstrate that while `EZNAS-(R-C-B)-(Mean)` is more effective, we are able to discover ZC-NASMs with all three formulations.

## 1.7 Hyper-parameters for discovering `EZNAS-A`

```
num_generation:  15
population_size:  50
tournament_size:  4
MU: 25
lambda_ :  50
crossover_prob:  0.4
mutation_prob:  0.4
min_tree_depth:  2
max_tree_depth:  10
```

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

| Op ID | Operation | Description
Output: `C`, Input: `A, B` |
|---|---|---|
| OP0 | Element-wise Sum | `C = A+B` |
| OP1 | Element-wise Difference | `C = A-B` |
| OP2 | Element-wise Product | `C = A*B` |
| OP3 | Matrix Multiplication | `C = A@B` |
| OP4 | Lesser Than | `C = (A<B).bool()` |
| OP5 | Greater Than | `C = (A>B).bool()` |
| OP6 | Equal To | `C = (A==B).bool()` |
| OP7 | Log | `A[A<=0] = 1`
`C = torch.log(A)` |
| OP8 | AbsLog | `A[A==0] = 1`
`C = torch.log(torch.abs(A))` |
| OP9 | Abs | `C = torch.abs(A)` |
| OP10 | Power | `C = torch.pow(A, 2)` |
| OP11 | Exp | `C = torch.exp(A)` |
| OP12 | Normalize | `C = (A - A`$_{mean}$`)/A`$_{std}$
`C[C!=C] = 0` |
| OP13 | ReLU | `C = torch.functional.F.relu(A)` |
| OP14 | Sign | `C = torch.sign(A)` |
| OP15 | Heaviside | `C = torch.heaviside(A, values=[0])` |
| OP16 | Element-wise Invert | `C = 1/A` |
| OP17 | Frobenius Norm | `C = torch.norm(A, p='fro')` |
| OP18 | Determinant | `C = torch.det(A)` |
| OP19 | LogDeterminant | `C = torch.logdet(A)`
`C[C!=C]=0` |
| OP20 | SymEigRatio | `A = A.reshape(A.shape[0], -1)`
`A = A@A.T`
`A = A + A.T`
`e,v = torch.symeig(A, eigenvectors=False)`
`C = e[-1]/e[0]` |
| OP21 | EigRatio | `A = A.reshape(A.shape[0], -1)`
`A = torch.einsum('nc,mc->nm', [A,A])`
`e,v = torch.eig(A)`
`C = (e[-1]/e[0])[0]` |
| OP22 | Normalized Sum | `C = torch.sum(A)/A.numel()` |
| OP23 | L1 Norm | `torch.sum(abs(A))/A.numel()` |
| OP24 | Hamming Distance | `A = Heaviside(A)`
`B = Heaviside(B)`
`C = sum(A!=B)` |
| OP25 | KL Divergence | `C = torch.nn.KLDivLoss('batchmean')(A,B)` |
| OP26 | Cosine Similarity | `A = A.reshape(A.shape[0], -1)`
`B = B.reshape(B.shape[0], -1)`
`C = torch.nn.CosineSimilarity()(A, B)`
`C = torch.sum(C)` |
| OP27 | Softmax | `C = torch.functional.F.softmax(A)` |
| OP28 | Sigmoid | `C = torch.functional.F.sigmoid(A)` |
| OP29 | Ones Like | `C = torch.ones_like(A)` |
| OP30 | Zeros Like | `C = torch.zeros_like(A)` |
| OP31 | Greater Than Zero | `C = A>0` |
| OP32 | Less Than Zero | `C = A<0` |
| OP33 | Number Of Elements | `C = torch.Tensor([A.numel()])` |

Table 1: List of operations available for the genetic program.