# OpenReview forum: "EZNAS: Evolving Zero-Cost Proxies For Neural Architecture Scoring"
_NeurIPS.cc/2022/Conference — NeurIPS 2022 Accept_

### Official Review · Reviewer_UZds · 2022-07-10

**Rating:** 6
**Confidence:** 4
**Soundness:** 3 good
**Presentation:** 2 fair
**Contribution:** 3 good

**Summary:**

This work proposes a framework, namely EZNAS, searching for zero-cost proxies based on evolutionary algorithm. Specifically, the authors utilize an expression tree to represent the computation of a zero-cost proxy and adopt DEAP evolutionary algorithm framework. Extensive experiments on NAS benchmarks show the effectiveness of the method.

**Questions:**

1. How do you obtain the accuracy of architectures sampled from NDS (Line 253-256)? Do you train theses networks from scratch or directly use the accuracy in NAS-Bench-301?

**Limitations:**

The limitations are addressed.

**Strengths And Weaknesses:**

Strengths:
+ The motivation is interesting and sound. This work attempts to search an effective zero-cost proxy rather than manually design one like prior works.
+ The experimental results are pretty good.

Weaknesses:
- The expression tree representation and the evolutionary algorithm is not new, which slightly limits the novelty.
- The layout is informal.
a) Fig. 1 should be placed behind the Fig. 2.
b) I suggest the authors providing a detailed legend in Fig. 2 to explain the meanings of T1(G), T2(G), T3(G), and T4(G).
c) Figure 5 should be Table 5.

---

> ### Author Response · Authors · 2022-07-28
> **Response to Reviewer UZds**
>
> We thank the reviewer for the positive feedback on our work.
>
> While it is true that the expression tree representation and evolutionary algorithm is not entirely novel (as we utilize principles of genetic programming to realize the search framework), we strongly believe that the application of the previously existing concept to a new domain, as well as the demonstrated efficiency of the method in discovering programs that are state of the art on unseen data-sets is extremely useful for the research community. We further detail some of these points in the common response as well as in `Response to Reviewer anAK` and request the reviewer to refer to it.
>
> Our representation and evolutionary algorithm design choices are a reflection of several cycles of non-trivial trial and error, and we believe that the methodology we have introduced in our paper is superior to alternative traditional methods. For instance, our attempts at using integer representations of programs led to failures in functional equivalence checking as well as program length bloating. This is explained in deeper detail in the Appendix of the paper.
>
> We would like to thank the reviewer for pointing out the Figure 1 and Figure 2 placement inconsistency, as well as the Figure / Table mix-up in Figure 5. We will attempt to include the meanings of T1(G), T2(G), T3(G), and T4(G) as a legend in Figure 2 in the final version of the paper.
>
> `How do you obtain the accuracy of architectures sampled from NDS (Line 253-256)? Do you train theses networks from scratch or directly use the accuracy in NAS-Bench-301?`
>
> We do not train the NDS networks from scratch. We directly use the accuracy from the NDS API (A fixed data-set).

---

> > ### Comment · Reviewer_UZds · 2022-08-08
> > **Thank the author for the feedback**
> >
> > I appreciate the authors' effort in the rebuttal. I decide to keep my original rating.

---

### Official Review · Reviewer_DDGZ · 2022-07-11

**Rating:** 6
**Confidence:** 4
**Soundness:** 3 good
**Presentation:** 4 excellent
**Contribution:** 3 good

**Summary:**

This work concentrates on crafting training-free measurement for neural architecture at initialization. Instead of being designed by human experts, the authors propose a genetic programming framework to automate the discovery of zero-cost proxies. Each candidate is represented by a computational graph of operation trees. The initial population is randomly constructed. At each iteration, an evolution task subset is sampled from the full datasets to evaluate, and the candidates with higher ranking correlation with the final performance is preserved. Experiments on a series of NAS benchmarks shows that the searched zero-cost proxies are more robust than the manually designed scoring functions.

**Questions:**

What does the searched EZNAS-A and EZNAS-B tree look like? I think it is important to visualize them or comprare them with typical zero-shot proxies.

**Limitations:**

No limitation is discussed in the paper.

**Strengths And Weaknesses:**

# Strengths:
1. I believe that this paper provides an insightful idea for the AutoML community that we can adopt evolutionary search on computational graphs for a specific operation with infinite searching space. Zero shot NAS is one example.
2. Experiments show that the searched proxy can generalize to different datasets, with extremely small searching cost (a single CPU in under 24 hours)
# Weaknesses:
1. Actually, this paper can be regarded as a new predictor-based NAS method, where the predictor in this work is searched by genetic algorithm. A general framework for predictor-based NAS is to construct a surrogate predictor (e.g. a meta network) to predict the final performance. However, none prior work on predictor-based NAS  has been referenced in this paper, nor does the author discuss its difference with them. I suggest the author should include them in your discussion.
2. Some of the very important baselines are missing. There exists a new trend for zero-cost NAS on neural tangent kernels[1, 2], which is cited in the paper but not presented in the quantitative comparison. I highly recommend the authors to compare with them.
3. The search space is quite restricted, "ReLU-Conv2D-BatchNorm2D", which hinders its application to more advanced architecture nowadays.
[1] Shu Y, Cai S, Dai Z, et al. NASI: Label-and Data-agnostic Neural Architecture Search at Initialization[J]. arXiv preprint arXiv:2109.00817, 2021.
[2] Chen W, Gong X, Wang Z. Neural architecture search on imagenet in four gpu hours: A theoretically inspired perspective[J]. arXiv preprint arXiv:2102.11535, 2021.

---

> ### Author Response · Authors · 2022-07-28
> **Response to Reviewer DDGZ**
>
> We would like to thank the reviewer for identifying the value of the idea for the community as well as the note on the efficiency of our method. We would also like to thank the reviewer for their thorough review.
>
> `Actually, this paper can be regarded as a new predictor-based NAS method....`
>
> To the best of our knowledge, ours is the first paper that automates the search for predictors in predictor based NAS. In this paper, we focus on introducing a novel methodology to discover programs for training-free measurement of accuracy of neural network architectures. We have included and compared with several relevant works which propose new training-free accuracy predictors on NDS, NATSBench and NASBench-201.
>
> We do not focus on integrating our programs with NAS, as that would further warrant an in-depth study of the effectiveness of the heuristic search methods that utilise the ZC-NASM. This would be out of scope for our paper due to space constraints. We would like to inform the reviewer that we have integrated EZNAS-A with a simple Aging Evolution search adapted from [3] in Figure 4 to demonstrate that our ZC-NASMs can be used for this purpose as well.
>
> We will definitely include appropriate references for predictor based NAS research in the related works as well as the discussion in the final version of the paper to emphasize on the differences between them.
>
> `Some of the very important baselines are missing.`
>
> TE-NAS [2] does not provide score-accuracy correlation metrics on most exisiting NAS datasets. Thus, we have intentionally cited but not included TE-NAS in our results. The only correlation metric provided is -0.42 Kendall Tau on NAS-Bench201 CIFAR-100, we out-perform their metric significantly by delivering a Kendall Tau Rank Correlation of 0.65 on NASBench201 CIFAR-100.
>
> We thank the reviewer for bringing our attention to [1], which was accepted at ICLR a few months before the NeurIPS deadline. We will incorporate this reference in the final version of the paper.
>
> `The search space is quite restricted, "ReLU-Conv2D-BatchNorm2D"...`
>
> We would like to point out that the search space itself is not restricted, it is a limitation of the expression tree representation. There are architectures in the search space which do not obey the "ReLU-Conv2D-BatchNorm2D" convention (as discussed in the paper), but the evolutionary search is still able to find expression trees/ZC-NASMs that deliver state of the art results on the search space. We have provided a detailed account the existing limitations of our method in the discussion to motivate future work for the community.
>
> `What does the searched EZNAS-A and EZNAS-B tree look like? I think it is important to visualize them or comprare them with typical zero-shot proxies.`
>
> The exact expression tree for EZNAS-A and EZNAS-B can be found in Figure 6 of the paper. We also provide an in-depth analysis of its behavior in Section 5.
>
> References:
>
> [1] Shu Y, Cai S, Dai Z, et al. NASI: Label-and Data-agnostic Neural Architecture Search at Initialization[J]. arXiv preprint arXiv:2109.00817, 2021.
>
> [2] Chen W, Gong X, Wang Z. Neural architecture search on imagenet in four gpu hours: A theoretically inspired perspective[J]. arXiv preprint arXiv:2102.11535, 2021.
>
> [3] Mohamed S. Abdelfattah, Abhinav Mehrotra, \Lukasz Dudziak, & Nicholas D. Lane (2021). Zero-Cost Proxies for Lightweight NAS. In International Conference on Learning Representations (ICLR).

---

> > ### Comment · Reviewer_DDGZ · 2022-08-07
> > **Thank the author for the feedback**
> >
> > I appreciate the authors' effort in the rebuttal. I decide to keep my original rating.

---

### Official Review · Reviewer_anAK · 2022-07-11

**Rating:** 6
**Confidence:** 3
**Soundness:** 3 good
**Presentation:** 3 good
**Contribution:** 3 good

**Summary:**

This paper automates the discovery of zero-cost proxies in NAS. Very low-level mathematical operations are selected and combined through genetic programming to form proper zero-cost proxies. The effectiveness of the proposed method is validated on NASBench-201 and NDS.

**Questions:**

None

**Limitations:**

Yes

**Strengths And Weaknesses:**

Strength:

1.	The paper is well motivated and clearly written. The paper is clear about limitations too.

2.	I think this paper will be valuable to the community. Zero Cost Proxies discovery is important for efficient neural architecture search. Evolving the proxies automatically from low-level mathematical operations also follows the trend of auto-ml.

3.	The searched zero Cost Proxy can better capture the order of candidate models than previous hand-crafted proxies, and is reported can generalize better on different search spaces.

Weakness:

1.	The proposed method looks quite complicated, considering the program representation, NN statistics generation as well as the evolutionary scheme. The inclusion of code would be a large benefit to this work.

2.	This paper borrows ideas from [1], providing the program synthesis is also considered before, which weakens the novelty of this paper. So I recommend a weak accept.

References:

E. Real et.al. Automl-zero: Evolving machine learning algorithms from scratch. ICML ‘20.

---

> ### Author Response · Authors · 2022-07-28
> **Response to Reviewer anAK**
>
> We thank the reviewer for their thorough review, as well as their positive comments on the motivation and value of the paper to the community.
>
> `The inclusion of code would be a large benefit to this work.`
>
> We plan to open source the code as part of a larger framework at a later date. We have made every effort to describe in full detail the mathematical operations used, network statistics collection strategy as well as the exact search hyperparameters required for discovering EZNAS-A in the Appendix.
>
> `This paper borrows ideas from [1], providing the program synthesis is also considered before, which weakens the novelty of this paper. So I recommend a weak accept.`
>
> We would like to thank the reviewer for bringing up such a relevant article. However, the idea of program synthesis has existed long before [1]. Our paper is inspired by some of the ideas introduced in [1] and we have appropriately cited the same, however we would like to clarify that both our problem formulation and use-case are extremely different. The program representation given in [1] fails at discovering effective ZC-NASMs in our design space, due to program length bloating and difficulty of efficient functional equivalence checking. We strongly believe that our formulation is valuable to the community for this distinct and specific problem, and requires non-trivial empirical testing to establish. We have detailed the inefficiences of using the approach introduced in [1] in further detail in the Appendix in Section A 1.2.
>
> References:
>
> [1] E. Real et.al. Automl-zero: Evolving machine learning algorithms from scratch. ICML ‘20.

---

### Official Review · Reviewer_gsE5 · 2022-07-12

**Rating:** 5
**Confidence:** 3
**Soundness:** 2 fair
**Presentation:** 2 fair
**Contribution:** 3 good

**Summary:**

Although NAS automates the manual design process of networks, it still requires expensive computing costs for evaluating multiple models. Zero cost proxies have been studied for alleviating the issues but it has several limitations, low productivity and poor ability of generalization. To tackle the current limitation of it, this paper introduces a genetic programming driven methodology, EZNAS, to automatically discover Zero-Cost Neural Architecture Scoring Metrics (ZC-NASMs)  that are interpretable, generalizable and deliver state of the art score-accuracy correlation.


**Questions:**

I can only see the score-accuracy correlation in the main paper and the supplementary documents.

If possible, please provide actual test accuracy and statistics, and detailed information, (i.e. training times, search times, FLOPs, model sizes, etc) of sampled networks for Table 1-5.


**Limitations:**

No negative societal impact is expected.

**Strengths And Weaknesses:**

Clarity: This paper is well-organized and easy-to-read. The authors smoothly develop their logic and key concepts throughout the paper. The concept figures are illustrated clearly.

Quality: This paper smoothly develops its logic throughout the paper. The experiments seem carefully designed.

Originality: The contribution of this work seems solid, since unlike the existing methods that use ZC-NASMs that are driven by human intuition or theoretical inspiration, this work devises a method for automatically discovering ZC-NASMs which lead to state-of-the-art score-accuracy correlations. This work has the potential to reduce human bias in the design of ZC-NASMs.

Significance: The achievement of this work seems significant, sinc it shows better score-accuracy correlation compared to the existing metrics and methods, i.e. DARTS, Amoeba, etc.

---

> ### Author Response · Authors · 2022-07-28
> **Response to Reviewer gsE5**
>
> We thank the reviewer for identifying that the study of zero cost proxies has been riddled with low productivity and lack of generalisation. We would like to thank the reviewer for their kind comments on the quality and originality of the paper.
>
> `I can only see the score-accuracy correlation in the main paper and the supplementary documents.`
>
> The score-accuracy correlation is generated by enumerating every neural network architecture in the search space (over 15000 networks on each of the NASBench-201 datasets and 5000 networks on each of the NDS spaces), and is a very strong metric to quantify the quality of a zero shot neural architecture ranking program. In an effort to focus on the methodology used to discover ZC-NASMs and describe its limitations, we do not focus on integrating the ZC-NASMs with existing state of the art heuristic neural architecture search algorithms. We believe that warrants a separate manuscript for two reasons. Firstly, our ZC-NASMs are designed to maximize the score-accuracy correlation as a fitness, and not 'search speedup (Figure 4)'. Secondly, extensive studies on different heuristic NAS methods that utilize our ZC-NASM would have to be described and tested for. Such an in-depth study which focuses on heuristic neural architecture search algorithms would be out of scope for a 9 page paper which introduces a novel method to discover ZC-NASMs. Nevertheless, we have integrated EZNAS-A with a simple Aging Evolution search adapted from [1] in Figure 4 to show that our ZC-NASMs can be used for this purpose.
>
>
> `If possible, please provide actual test accuracy and statistics, and detailed information, (i.e. training times, search times, FLOPs, model sizes, etc) of sampled networks for Table 1-5.`
>
> As this is a neural architecture ranking program, we use it to rank every neural network in the design space. We sample all neural networks in the search space, and thus cannot feasibly provide their detailed information in this paper. These details are contained in the respective paper associated with the data-set. Further, we feel that these details are not required in our work to prove the effectiveness of the ZC-NASM in ranking programs. We may include such details in a future manuscript where we use the discovered ZC-NASMs with heuristic search algorithms for studying the search efficiency.
>
>
> [1] Mohamed S. Abdelfattah, Abhinav Mehrotra, \Lukasz Dudziak, & Nicholas D. Lane (2021). Zero-Cost Proxies for Lightweight NAS. In International Conference on Learning Representations (ICLR).

---

> > ### Comment · Reviewer_gsE5 · 2022-08-09
> > **Thank you for the response.**
> >
> > I thank the author for providing detailed responses. My curiosity has been addressed.  Thank you.

---

### Author Response · Authors · 2022-07-28
**Common Response**


We would like to thank the reviewers for the thorough and insightful assessment of our paper.

In this paper, we take the first steps towards a framework that can reduce the bias and human effort required in zero shot neural architecture scoring. This is a very relevant automation problem for the Auto-ML community, as more neural architecture data-sets become public.

Extracting important design suggestions (as done in our paper in Section 5) and discovering efficient accuracy proxies are very relevant as neural architecture design problems abstract itself to increasingly complex multi-objective problems (latency, throughput, accuracy). Our method is able to discover ZC-NASMs that are state of the art in over 7 distinct, unseen neural architecture design spaces (Our EZNAS-A ZC-NASM searched on NDS DARTS is SoTA on all NDS and NASBench-201 design spaces) with minimal human intervention and less than 1 CPU-day worth of time and carbon emissions.

We believe that there is significant scope for future work in this field, as the program representation, data-collection and evolutionary search formulation can be modified to suit different use-cases and novel architectural descriptions. Thus, we have heavily focused on introducing our ZC-NASM search formulation as well as enumerating every neural network architecture in the design space on NASBench and NDS (over 71875 neural networks) to provide a solid, reproducable baseline (the ranking correlation over an entire neural architecture design space.).

---

### Meta-Review · Area_Chair_CqSe · 2022-08-26

**Recommendation:** Accept
**Confidence:** Less certain

**Metareview:**

This paper proposes a genetic algorithm for discovering zero-cost scoring metrics for neural architectures, that can generalize to neural networks in unseen search spaces. The proposed genetic algorithm based on the DEAP evolutionary algorithm can discover programs in the form of expression trees to represent the computation of a zero-cost proxy, and thus is also interpretable. The proxy found using the proposed algorithm is validated on the benchmark datasets such as NASBench-201, and is shown to outperform previous methods in the estimation of architecture’s accuracy, in terms of the Spearman’s correlation score.

All reviewers gave the paper positive ratings, due to the importance of discovering zero-cost proxy for NAS, its superior performance estimation ability over existing proxies, and its quality of writing. However, reviewers were also concerned about possibly limited practical impact due to the complicated methods, lack of methodological novelty as it could be considered as a straightforward application of an existing genetic algorithm, and missing comparison against predictor-based NAS methods as well as state-of-the-art methods.

I also agree that while the proposed work could be a valuable addition to researchers working on zero-cost NAS, the missing discussion and comparison against relevant baselines such as predictor-based NAS is the main drawback of the method. In addition to methods that are based on Neural Tangent Kernels mentioned by Reviewer DDGZ, meta-NAS methods [Lee et al. 21] [Jeong et al. 21] that can generalize to new datasets should be also further discussed and compared against, considering both the performance estimation accuracy and training efficiency. Lack of the results and analyses of end-to-end NAS results is another drawback.

The author responses do not provide experimental results on these points that are raised by the reviewers, perhaps due to good initial scores. However, these missing discussions and comparisons makes it difficult to understand the pros and cons of the proposed work over theirs, which makes it a borderline paper at its current state. I strongly suggest the authors discuss existing works and what advantage the proposed method has over theirs (e.g. generalization to new search spaces, explainability) as well as perform experimental comparison over theirs.

[Lee et al. 21] Rapid Neural Architecture Search by Learning to Generate Architectures from Datasets, ICLR 2021
[Jeong et al. 21] Task-Adaptive Neural Network Search with Meta-Contrastive Learning, NeurIPS ICML.

**Award:**

No

---

### Decision · Program_Chairs · 2022-09-14

Accept